# Portable Arduino-Based Multi-Sensor Device (SBEDAD): Measuring the Built Environment in Street Cycling Spaces

**DOI:** 10.3390/s24103096

**Published:** 2024-05-13

**Authors:** Chuanwen Luo, Linyuan Hui, Zikun Shang, Chenlong Wang, Mingyu Jin, Xiaobo Wang, Ning Li

**Affiliations:** 1Department of Architecture, School of Architecture and Art, North China University of China, Jinyuanzhuang Road 5, Shijingshan District, Beijing 100144, China; huilinyuan@mail.ncut.edu.cn (L.H.); shangzikun@mail.ncut.edu.cn (Z.S.); rain_wcl@163.com (C.W.); jinmingyu@ncut.edu.cn (M.J.); wangxiaobo@ncut.edu.cn (X.W.); 2Beijing Historical Building Protection Engineering Technology Research Center, Beijing University of Technology, Beijing 100124, China

**Keywords:** built environment measurement, sensor integration, cycling infrastructure, smart city technology

## Abstract

The built environment’s impact on human activities has been a hot issue in urban research. Compared to motorized spaces, the built environment of pedestrian and cycling street spaces dramatically influences people’s travel experience and travel mode choice. The streets’ built environment data play a vital role in urban design and management. However, the multi-source, heterogeneous, and massive data acquisition methods and tools for the built environment have become obstacles for urban design and management. To better realize the data acquisition and for deeper understanding of the urban built environment, this study develops a new portable, low-cost Arduino-based multi-sensor array integrated into a single portable unit for built environment measurements of street cycling spaces. The system consists of five sensors and an Arduino Mega board, aimed at measuring the characteristics of the street cycling space. It takes air quality, human sensation, road quality, and greenery as the detection objects. An integrated particulate matter laser sensor, a light intensity sensor, a temperature and humidity sensor, noise sensors, and an 8K panoramic camera are used for multi-source data acquisition in the street. The device has a mobile power supply display and a secure digital card to improve its portability. The study took Beijing as a sample case. A total of 127.97 G of video data and 4794 Kb of txt records were acquired in 36 working hours using the street built environment data acquisition device. The efficiency rose to 8474.21% compared to last year. As an alternative to conventional hardware used for this similar purpose, the device avoids the need to carry multiple types and models of sensing devices, making it possible to target multi-sensor data-based street built environment research. Second, the device’s power and storage capabilities make it portable, independent, and scalable, accelerating self-motivated development. Third, it dramatically reduces the cost. The device provides a methodological and technological basis for conceptualizing new research scenarios and potential applications.

## 1. Introduction

Urban and architectural design processes are increasingly using built environmental data acquisition to support and optimize design outcomes. The design of fast-response, autonomous, and affordable mobile urban sensing technologies plays a significant role in acquiring high-spatial-resolution environmental data for urban planning [1]. By integrating spatial data and innovative technologies (e.g., geographic information systems, remote sensing, and the Internet of Things (IoT)), urban planners can gather valuable insights to design environmentally friendly indoor and outdoor spaces that are responsive to the needs of urban development [2,3,4]. Whether it is monitoring environmental parameters, assessing urban vitality, evaluating the function of urban furniture, or analyzing the impact of smart city planning, data collection plays a crucial role in urban design [5,6].

The term *built environment* refers to the human-made surroundings that provide the setting for human activity. It comprises urban design, land use, and the transportation system and encompasses patterns of human activity within the physical environment [7,8]. Data related to the built environment commonly used in planning and urban design are often macro or meso scale data, such as topography and geomorphology, weather, and points of interest (POI). These data are authoritative in macro-recognition. However, in mesoscale urban design, landscape design, and other urban open space designs, it is not possible to integrate multiple sources of data to achieve a deeper understanding of the city and provide a database for renewal construction. Therefore, there is an urgent need to supplement the fine granularity of data at the street level to promote a better understanding of the city by the designers and managers. The interplay between the built environment and group behavior can be fundamentally recognized based on a deep understanding of the microenvironment. Fine-grained urban open space data can help designers think about the spatial layout and connection of different functional areas in a neighborhood. Designers can then better consider the effect of urban open space on the shading of the surrounding buildings, wind, and the comfort and safety of pedestrians, which will help them improve the comfort and convenience of urban open spaces, and create a pleasant urban open space.

By combining advanced technologies such as multi-source data, 3D modeling, indoor positioning, and image recognition, designers can assess outdoor comfort, air quality, and microclimate to create more sustainable and livable environments. Support is also provided for environmental pollution, emergency response, flood risk management, and urban landscape design [9,10,11]. In addition, integrating data-driven approaches such as machine learning and big data analytics can provide valuable insights for optimizing urban planning and design processes. Using data from various sources and applying computational techniques, urban designers can evaluate design parameters, assess ventilation potential, and improve the economic and social benefits of urban spaces [12,13,14,15]. Currently, digital research related to urban design is mainly carried out by applying computer vision (CV) [16,17] and street view images (SVI) combined with machine learning, e.g., by analyzing SVI to understand landscape features such as buildings, streets, and parks in the city [18], which can help urban planners better understand the layout and structure of the city and provide the necessary data support for urban planning. Transportation planning can be optimized by analyzing traffic flow and pedestrian activity to improve efficiency and enhance pedestrian safety [19]. The city’s degree of greening and environmental quality can be assessed by reviewing green spaces and tree cover to provide data support for sustainable urban development [20]. Machine learning can be used to predict urban growth patterns and help formulate sustainable urban development strategies [21]. In the direction of research related to hardware sensing, Zhang collected data in real-time by deploying noise sensors in the city to provide an essential reference for improving the urban noise environment [22]. Based on a single intelligent sensor, the researchers developed a research methodology for data collection, analysis, feature extraction, machine learning prediction, and optimization decision-making, which brought an important impetus to sustainable urban development.

With the rapid development of artificial intelligence, machine learning, and deep learning technologies, the ability to process and analyze massive multi-source image and sensor data has also been greatly improved [23,24]. Data are the premise and foundation for machine learning and deep learning. At present, temperature, humidity, light, particulate matter (PM), noise, greenery, and other data from urban open spaces are seldom fused and collected for application. One reason for this is that there is simply no portable device that can ultimately collect the above data. Although some professional devices are single-sized and portable, there are no multi-source sensor devices designed for street-level urban data platforms. Realizing data collection in the same period will require more than five devices, and it must be ensured that the sensing ends of the devices are in adjacent spaces. If it is a split collection of different devices, it will lose its scientific nature due to different times and environments, and the problem of carrying a large number of devices and power consumption will also make the collection challenging to realize. For example, in understanding and measuring parks or streets in walking or cycling environments, motorized vehicles carrying large, complex multiple sensors cannot be used because they cannot be driven in a particular area; portable equipment is thus required. At the same time, understanding urban open space does not require the same stringent data collection accuracy and error requirements as laboratory research. In addition, cost reduction is a major consideration for small- and medium-sized companies or research departments.

The current Internet of Things (IoT) sensors have been applied to environmental detection and understanding in building and urban design. It can be divided into perception, transmission, and application networks according to their functions. SVI, data acquisition using sensors, and data fusion applications are the keys to realizing street-scale understanding and prediction of the urban built environment. Therefore, there is an urgent need to build a street built environment data platform (Figure 1).

This study designs, develops, and makes a low-cost, portable, multi-channel street built environment data acquisition device (SBEDAD) to provide a data platform for city managers and designers for designing, recognizing, and building a smart city.

## 2. System Requirements and Design

Urban monitoring data collection through a large number of sensors can support innovative applications for a variety of scenarios, such as environmental monitoring, security, health, education, and urban mobility [25]. Multi-sensors can detect urban land cover changes and surface urban heat islands (SUHI) within cities [26,27,28]. Vibration and deformation sensors can perform structural integrity measurements for historic buildings [29]. However, distributing and maintaining sensors in metropolitan areas is a challenge due to technological limitations [25,30]. The use of mobile sensors can significantly reduce the need to distribute static sensors to cover most urban areas for urban management and monitoring [31] and increase the level of spatial and temporal granularity [32], especially when using a cloud-based platform for crowd sensing in urban scenarios [33]. Detecting load levels and optimizing truck routes using intelligent monitoring sensors reduces the cost of urban waste management [34,35]. Sensors have been placed in vehicles, bicycles, or backpacks of pedestrians for opportunistic surveillance. Meteorological data and built environment data have been collected from mobile monitoring sensors and stationary sites to expand the spatial extent and temporal resolution [20,36,37,38]. Overall, there are various innovative approaches and technologies for monitoring urban environments efficiently and effectively.

Based on existing research, we realized that urban sensing research has a great need for first-hand, sensor-based mobile data collection. However, current research struggles to obtain urban stationery sensing data, which are not sufficiently dense. Developing or customizing the required equipment is both expensive and time-consuming, as it requires interdisciplinary knowledge. Last but not least, the type of sensing data cannot be changed to suit the needs of the study.

Arduino is a simple and easy-to-use open platform; it includes an open-source hardware and software IDE (version 2.2.1), providing unlimited possibilities for developers. It is developer-friendly and suitable for designers, other non-professional programmers, and non-professional embedded system developers [39]. The research team chose this platform to design and develop a low-cost portable Arduino-based street built environment data acquisition device (SBEDAD). It can not only measure the basic functions needed for this research but also add intelligent sensing components to the device as needed in future research and development processes to prepare for the research and development of urban open space.

### 2.1. Basis of the System Design

The system is based on the basic core of a microcontroller released by the open hardware Arduino Mega 2560 (Arduino, Turin, Italy). Mega 2560 is a microcontroller application development board printed circuit board (PCB) with an advanced control system (AVR) microcontroller as the core controller, and its ecosystem also provides a large number of application libraries to avoid complex register manipulations. Programming is conducted through programming languages derived from C and C++ to enable different sensors by powering the digital and analog pins on Mega 2560. (See Table 1).

The SBEDAD designed in this research aims to provide a data platform for future urban design and macro urban environmental studies. It integrates feedback on cycling and walking physical sensations, road quality, and greening quality and is fitted with several sensors, including PM concentration, luminous intensity, temperature, humidity, and noise sensors, as well as an 8K panoramic camera. The Mega board has sufficient pins, such as 54 digital input/output pins (of which 15 can be used as PWM outputs), 16 analog inputs, and 4 universal asynchronous receiver/transmitter (UART) hardware serial ports [40]. The GPS and PM sensors, which communicate based on the URAT protocol, are connected to the serial port. The temperature and humidity sensors, which are based on the transistor–transistor logic (TTL) protocol, are connected to the digital pin, while the light intensity sensor and noise sensor are connected to the analog port. The 3.2-inch LCD screen and SD card module, which are based on the serial peripheral interface (SPI) protocol, are connected to the SPI pins. To ensure fast and correct data transfer from sensors using different protocols and ports, we added an expansion board and a module in the script to check the flash availability before recording sensing data. Table 2 summarizes the configuration of the entire device. The circuit system was ultimately mounted in a specially designed enclosure to facilitate assembly and disassembly. The enclosure was made using a Bambu P1P machine, 3D printed in PLA material and secured with drop-in glue and screws. (See Figure 2).

Currently, sample intervals in urban open spaces are usually two to five seconds [20]. Data acquisition for urban open spaces is a unidirectional sensing data collection process rather than an interaction process. The precision of temperature and humidity, air quality, light, noise, and other to-be-measured data in the urban built environment, which are used to study the impact of the urban environment on human behavior, is usually required at the level of seconds. Data with ultra-high precision merely have an impact on the study objectives. It should be emphasized that the ADC of the Arduino board is of 10-bit resolution and measures 0 to 5 V. This 10-bit resolution allows us to achieve an accuracy of ±4.882 mV (beyond the technical requirements). Moreover, to ensure that the user can use the device in daily environments and that the device works properly, the device provides the user with a screen, a secure digital (SD) card for direct storage of data, a power supply, and a reboot switch.

In configuring the parameters for SBEDAD, careful consideration was given to the specific values of the various setup options to meet the characteristics of the target use case and optimize system performance.

Based on the Air Quality Guidelines (AQG) issued by the World Health Organization (WHO) in 2001, standard values were set for a wide range of air pollutants, such as PM2.5 and O_3_. The guidelines provide optional milestones for air quality management in countries around the world. The PM2.5 concentrations and PM10 concentration thresholds were set to 15 and 45 μg/m^3^ [41]. These data can help complete an exposure risk assessment of pollutant concentrations.

Temperature and humidity are two important meteorological elements that determine physical comfort. The human body has a certain ability to adapt to changes in external temperature and can maintain balance with the help of thermoregulation. However, the body’s regulation has a certain limit, in which the upper limit temperature is 32 °C and the appropriate relative humidity is 30% to 80% [42].

The perceived quality of street lighting affects cyclists and pedestrians’ sense of safety and visual comfort, as well as outdoor activities at night. Existing research has revealed that illumination at 5 to 17 lx is a satisfactory level, and people feel safer if the light is warm, even at night [43].

High levels of noise can lead to physical and psychological problems for residents. Noise in the street space has a severe impact, especially on cyclists and pedestrians, and the construction of highly dynamic noise data streams can be used to assess and effectively manage traffic noise pollution [20].

Based on existing research, the data on PM, temperature, humidity, light, and noise in the street space are factors that have a greater impact on the comfort and experience of human beings directly exposed to the street space. Therefore, the data collection scope of SBEDAD is based on these data, and the sensors can be simply replaced to obtain different data according to the needs of different urban designers and managers in the future.

### 2.2. Low Cost

The manufacturing budget for SBEDAD is broken down into different components in Table 3. The total budget for the obligatory part of the basic version of the project is USD 68.78. The cost of adding other extended functionality parts will increase with the price of the required components, but it will not exceed USD 150.

### 2.3. Open-Source Software

The hardware of the device was programmed using an integrated development platform (IDE) that is compatible with different platforms (Linux, Windows, and Mac) and is available free of charge under the GNU General Public License. The programming language used for the device was derived from C and C++, and the program flowchart is shown below (Figure 3).

To connect multi-source sensors, this study built an initial prototype using an extension board to connect the sensors to the Mega board for multi-channel power supply and receiving data. The extension board allows easy changes in wiring and connections, exchange of components, and software testing until the desired functionality is achieved. After the sensor function tests are completed one by one, the PCB board and all its components are encapsulated in a custom-made 3D-printed thermoplastic case of PLA material. Power is provided by a rechargeable 5–12 V lithium battery connected to the external power socket of the Arduino board. Figure 4 shows the SBEDAD structure.

## 3. Functional Validation

We undertook a comprehensive functional validation of the multi-source data acquisition capabilities of the street built environment data acquisition device (SBEDAD), underpinned by environmental sensing theories and methodologies for urban data analysis. Commencing with the environmental protection standards for air pollution monitoring, we integrated models of human thermal comfort, frameworks for evaluating road conditions, and ecological impact theories of urban greening, thereby systematically analyzing and substantiating the validity and reliability of the data procured by SBEDAD. This approach thoroughly examined SBEDAD’s performance across multiple dimensions of the urban built environment.

Following the established criteria for evaluating green spaces, we employed image data captured by an 8K panoramic camera to calculate vegetation coverage rates, utilizing an algorithm specifically designed for this purpose. A comparative analysis was conducted against empirical survey data (Table 4), revealing a correlation coefficient of r = 0.87 between the two datasets. This finding attests to a high degree of consistency in SBEDAD’s assessment of urban green spaces, thereby validating its efficacy in quantifying and characterizing green infrastructure within urban environments.

Through experimentation and analysis, the accuracy and error of each sensor used in SBEDAD were determined, as well as the corrections required for the final readings of the device. It should be noted that the device lost some of its accuracy requirements to achieve portability compared to the high accuracy of industrial equipment. The reason for this is that, for the accuracy of sensory data acquisition in urban open spaces, such as relating to temperature, the perceptual threshold of the body’s somatosensory senses in outdoor environments is approximately 1 °C [44,45]. Based on existing research, the required accuracy of SBEDAD was analyzed using empirical values, and finally, the accuracy of the data acquired by each sensor of SBEDAD was determined (Table 4).

According to the existing research results, the temperature data acquisition precision was set to one decimal place, i.e., 0.1 °C ± 10%. Humidity acquisition accuracy was 0.1% ± 10%; light intensity was 100 lux ± 10%. The wind speed accuracy was 5 m/s ± 0.1 m/s. Each value of PM was 2 μg/m^3^ ± 10%.

Two different evaluation calibration activities were performed to confirm the accuracy of different sensors and compare the sensors’ readings directly with professional industrial instruments and equipment, including temperature, humidity, light, PM, and noise sensors for acquiring values. These tests represent practical examples of the use of the Arduino-based development of SBEDAD in a well-established technical program. These experiments were carried out in normal temperatures between −10 and 45 °C, with humidity below 80%, to represent the real acquisition environment in urban open spaces instead of considering calibration residuals in the extreme temperature and humidity environment of the laboratory. (See Figure 5).

### 3.1. GPS and BDS Longitude and Latitude

The SBEDAD is designed to support urban design and city management, so the data it acquires should have temporal and spatial characteristics. By fusing data on temperature, humidity, air pollution, light, and noise with geographic location and time correspondence, it is possible to obtain mutually reinforcing knowledge from data sources with different attributes and origins and ultimately establish correlations between different data. (See Table 5).

The SBEDAD is equipped with two sets of latitude and longitude positioning sensors, namely, the global positioning system (GPS) and the BeiDou system (BDS), for determining the location geographically. At the same time, the coordinated universal time (UTC) is taken as a marker to unify with the data from the panoramic camera and then to fuse and analyze the spatial–temporal characteristics of the urban open space. We programmed the Arduino IDE (Figure 6a) to obtain real-time GPS and BDS information, including UTC, latitude, and longitude, through serial communication. During the research process, we also used a set of Pilot One 8K panoramic video cameras, which have a built-in GPS sensor, to collect video data. Eventually, this study carried out calibrations on the Baidu map by comparing it with GPS (ATGMH-5N) and BDS data.

It should be noted that the ATGM336H-5N module is an electrostatic-sensitive device. Constant electrostatic contact can cause unexpected module damage. It should connect the GND pin first and then the signal cable before use. An oscilloscope was also used to read the real frequency of the ATGM336H-5N module, which was 100 milliseconds (Figure 6b), much less than the sampling frequency required by the SBEDAD (1 s, as mentioned earlier).

We conducted several field data acquisition experiments for GPS and BDS sensors and finally decided to use Mega’s serial port 3 for ATGM336H-5N data reading. We compared the data acquired by this sensor in Beijing, a city in North China whose temperatures range from −11 to 6 °C with an average temperature of 2 °C, in January 2024, and in Sanya, the southernmost city of China, where the temperatures range from 14 to 29 °C with an average temperature of 22 °C, in January 2024. The accuracy of the GPS module was verified through field tests in different outdoor temperature and humidity environments as well as latitude and longitude geographic locations.

A set of positions were obtained through the built-in sensing elements in Beijing and Sanya. Encrypted location information was then translated and analyzed on the Baidu map (Figure 7), which is compatible with both GPS and BDS. The results showed that the latitude and longitude position deviation was within 2.5 m, which meets the accuracy of information acquisition for urban spatial information.

### 3.2. Environmental Temperature and Relative Humidity

The calibration of the temperature and humidity sensors was performed in a real office environment. The experiment compared the data collected by two sets of sensors (DHT11 and DHT22) and an industrial-grade commercial sensing device (UNI-T UT332+) (Table 6). The data acquired by the sensors was logged, read, and printed on the local side through the serial port. The UT332+ could log 100 sets of data in seconds, then read and print them through the host computer application (Figure 8a). The real frequency of the DHT22 obtained by the oscillograph was 10 milliseconds (Figure 8b).

To test out the accuracy of the temperature and humidity sensors more efficiently, the experiment used four environmental stimuli to compare the response sensitivity and accuracy (Figure 8c):Hairdryer heated blowing mode;Hairdryer cool blowing mode;Ice-water mode;Breathe warm air mode.

As shown in Figure 9, we collected 600 sets of data to compare the temperature and humidity on DHT11, DHT22, and UT332+ at the same time and in the same scenario. In the temperature-sensing performance, a percentage difference below 2% was recorded in 29.18% and 47.87% of cases, respectively. Most data fell within a percentage difference of 5%, with 64.89% and 80.85% of cases, respectively. In the humidity-sensing performance, a percentage difference below 2% was recorded in 38.30% and 58.82% of cases, respectively, and a difference below 5% was recorded in 72.19% and 78.82% of cases (Figure 9). In both temperature and humidity aspects, DHT22 showed more accurate sensing ability and stable environmental sensing ability than DHT11.

Through comparisons in the laboratory environment, SBEDAD finally selected the DHT22 as the final option for the temperature and humidity sensor.

### 3.3. Air Pollution of PM1, PM2.5, and PM10

The sensing data for air quality is mainly for particulate matter (PM) values, and sensors for CO, NO, NO_2_, SO_2_, and other gas detection can also be added in future research. The three most representative suspended particulate matter detections are PM1, PM2.5, and PM10 because they can be suspended in the air for a longer period, and the higher the concentration of their content in the air, the more serious the air pollution.

In this study, the Plantower G7003S laser particle sensor was selected, which uses the laser scattering principle to achieve accurate measurements with an extremely low false alarm rate and a minimum resolved particle size of 0.3 µm. (see Table 7).

By the oscilloscope test, the real frequency of the Plantower G7003S was 100 milliseconds (Figure 10a), which meets the SBEDAD requirements. Conventional PM measurements performed by gravimetric methods detect the change in mass of PM as it passes through a filter membrane. Plantower G7003S, on the other hand, uses the optical method of measurement, which has a short sampling period and allows for real-time monitoring, providing timely PM values. The technology is relatively simple and does not require complex equipment or operating procedures.

We used sandpaper to sand different materials, such as nails, plastics, and chalks, and then blew the particles away to generate different particulate matter concentrations in the environment. The accuracy of the Plantower G7003s was verified through experimental analyses. In the end, we chose the G7003S.

### 3.4. Illumination

The illumination on the road has a very strong and obvious impact on the cycling experience. SVIs, which have also been used to analyze shading and illumination, are mostly collected by the data sampling vehicle that traveled on the driveways and are usually more open to the driving view. This kind of SVI does not inherit the ambient light feature on the cycling path. The SBEDAD collects the value change of the illumination on the cycling road, assisted by the panoramic camera and the temperature and humidity sensing data, which can be used to prove the comfort of the cycling experience in multiple dimensions.

The calibration of ambient lighting was performed in two environments: laboratory and outdoor. This experiment compared the data collected by two low-cost sensors (TEMT6000 and BH1750) and an industrial-grade commercial sensing device (the Konica T-10A). (See Table 8).

Based on the experimental design, the TEMT6000, BH1750, and Konica T-10A were placed in a dimly lit room under a lamp as the light source. The illuminance was measured at a distance of 20 cm from the light source to ensure that the light cone had a large enough diameter and to ensure uniform conditions for the different sensors in the measurement plane.

In addition, the real-frequency measurements of both sensors were conducted with an oscillograph. The data showed 5 and 20 ms, respectively (Figure 11). Through experimental analysis, it was found that the analog value of TEMT6000 stayed around 950 when the light intensity exceeded 1800 lx, which showed great instability and inaccuracy. Therefore, we ruled out this sensor.

After determining the portable sensor, its accuracy needs to be checked and calibrated within the field test. We collected data on light in a building’s corridor, in the entrance hall, under the canopies, and in the outside open space around buildings (Figure 12b). We collected 100 sets of data in a daily light environment and fit the data from BH1750 to T-10A to perform the data calibration for BH1750.

The BH1750 data were fitted to the Konica T-10A data using both polynomial and exponential functions to calibrate the data taken by the BH1750 (Figure 13). The formulas are as follows:(1)$$y=1E-07x^{6}-4E-05x^{5}+0.0043x{\;}^{4}-0.2085x^{3}+4.733x^{2}-39.174x+131.47$$
(2)$$y=46.375\times e^{0.0386x}$$

By contrast, we chose Formula (2) as the prediction and calibration formula. There are two ways to use this equation correction: we can directly input the formula in the Arduino IDE for numerical correction, or we can correct it after obtaining the values.

### 3.5. Environmental Noise Intensity

Traffic noise occupies more than 60% of the urban acoustic environment [20]. It is an important factor in polluting urban transportation space. The noise made by vehicles has little effect on drivers but has a much more serious impact on cyclists and pedestrians. Therefore, it has been mentioned in several studies [20,49,50,51] that the noise in the street space is a very important aspect of studying the urban street spatial environment.

We compared sensors with stable performance in different temperature and humidity environments and finally chose Yahboom’s professional sound sensors (see Table 9), which are sensitive to ambient noise and have a range suitable for acquiring ambient noise in street spaces. Also, to improve the accuracy of the data, we calibrated the Yahboom data using the sensor inside the cell phone.

We used low-pass and high-pass filters with amplifiers to reduce the noise in this sound level measurement circuit so that the accuracy could be improved. Similarly, we used an oscilloscope to test the real frequency of the sensor, which was 10 ms (Figure 14).

The relationship between the decibel value (dB) and the simulated value is not linear, so we need to compare the two values at different intervals. We noted down 100 sets of the ADC value obtained by the Yahboom sensor and the dB value shown on the phone, then used the “linear regression” method to complete the data fitting. This converted the irregular blue scatter to the closest possible curve (red) and generated an equation for that curve. We conducted curve fit, 5th-degree polynomials fit, and 10th-degree polynomials fit using a Python script (Figure 15) and eventually chose the 10th-degree polynomials formula (Formula (3)) to deliver data fitting.
(3)$$dB=-1.179e-21\;x^{10}+2.569e-18\;x{\;}^{9}-2.38e-15\;x^{8}+1.237e-12\;x^{7}-4.013e-10\;x{\;}^{6}+8.554e-08\;x^{5}-1.216e-05\;x^{4}+0.001132\;x^{3}-0.06548\;x^{2}+2.14\;x+26.25$$

## 4. Field Measurement Results

The experiment finally selected a sample of street cycling space in central Beijing for data collection using SBEDAD, which proved to be efficient and accurate, showing good reliability through the continuity and changing characteristics of the data. The application can be expanded to work with other slow-walking spaces for multi-source data collection, providing a basis for the use of specialized sensors as well as a more critical approach to conceptualizing new scenarios and potential applications.

### 4.1. Field Data Acquisition Process and Results

The study replicated two sets of equipment in two groups of students for a total of 12 data collection sessions over three days (36 h in total) and collected 127.97 G of video data and 4794 Kb of txt records. Tests were carried out on 15 streets in the center of Beijing to check the operation of the equipment under real-world conditions. Compared to the completion of field data collection of similar streets by students in the researcher’s design studio in the same period in the previous year, only 5.97 G of non-panoramic video data and 100 questionnaires were acquired by four research teams in 144 h during 12 days. Moreover, less than 50 questionnaires were valid. The working efficiency using SBEDAD rose to 8474.21% compared to last year. In addition, the streets’ GPS location records, temperature, humidity, concentration of PM, light intensity, and noise data were acquired simultaneously in the form of text. An increased variety and density of data provides comprehensive information on the functioning and changes in cities and can lead to an intrinsic change in data-based research. (See Figure 16).

The use of multi-source and multi-sensor data collection tools greatly improved the efficiency of street spatial analysis and research and provided a database and scientific basis for subsequent scientific research.

The research used both the panoramic camera and the SBEDAD (Figure 17). This device is convenient, portable, consumes low energy, and can be placed in the basket of a shared bicycle for data collection.

Information extraction from SVI obtained by a panoramic camera enables the prediction of urban perception. The development of computer vision (CV) ensures the automatic and efficient processing of large numbers of photographs. We used a Python script to frame-pump the video. Because SBEDAD data were sampled per second, one frame was extracted from the video captured by Pilot One, which was set up to 7 frames per second, and then fused with the SBEDAD data along with the timestamp. For instance, through semantic segmentation, a deep learning technology, we could automatically and efficiently obtain the greenery coverage of different street spaces, combined with the temperature and humidity data, which are directly affected by the greenery rate, to calculate and predict the impact of different built environments on the human thermal comfort sensation.

### 4.2. Fusing Data to Understand Cities

Using SBEDAD to acquire diverse and massive urban spatial data, i.e., realizing urban sensing and data acquisition, can help to understand the complexity of the city. The problems of data sparsity and missing data can also be solved by SBEDAD.

Based on SBEDAD, users, i.e., students of the researcher’s Urban Design Studio, can simply and easily acquire data and use data visualization tools to visualize spatial data (Figure 18) through the forms of colors, symbols, labels, etc., transforming abstract data into intuitive visual forms. Intuitively displaying data changes and trends in the city helps us better understand and analyze the spatial characteristics of the city and supports design, decision-making, and problem-solving in the city. (See Figure 19).

Sensing the city for effective planning and governance is critical to the development of the city. Simply interpreting a city through one type of data is probably not comprehensive. Understanding the city requires the integration of knowledge from multiple data sources. Interpreting data with a humanized and sustainable view of the city and objectively judging the state of the city’s operation are important. Ultimately, human feelings, consciousness, and social relationships can be improved, and data science can be used to simulate and deduce the effects of data-based urban optimization.

Multiple artificial intelligence algorithms and deep learning methods can be utilized to facilitate cross-domain data fusion in smart cities to capitalize on the intrinsic connections between data streams from different cities. Based on first-hand, multi-source, and cross-domain urban built environment spatial data, the combination of machine learning and deep learning techniques can promote the development of data fusion methods in the direction of urban computing.

## 5. Conclusions and Future Work

Multi-modal causal learning based on multi-source urban data acquired in this study aims to improve the interpretability of intricate and dynamic urban systems. GPS location, temperature, humidity, and light intensity data can explain the micro-environment of a specific urban space in detail. Data with temporal and spatial characteristics can be combined with artificial intelligence algorithms to learn and predict street spaces’ spatial and temporal characteristics.

The device based on the open-source Arduino platform presented in this study has proven to be a good choice for measuring built environment perception data in urban open spaces. The results obtained show that it is feasible to use a low-cost device for the measurement and recording of this type of data. In addition, the simplicity, portability, low energy consumption, room temperature stability, replicability, and low cost of the device compared to other commercial systems, as well as the possibility of scalability, adjustment, and improvement needed for future research, mean a reliable database can be obtained for city managers and program designers. Urban spatial analysis based on multi-source data therefore becomes possible. SBEDAD provides higher spatial resolution and area coverage for urban sensing and effectively addresses data inconsistencies arising from sensors and cameras.

This device has some limitations. Firstly, due to the short stay of the device, SBEDAD data has a high spatial resolution but a low temporal resolution, so machine learning and deep learning processing based on temporal features will be limited. Secondly, to ensure the data collected by SBEDAD had relatively high stability and reliability during data acquisition and transmission, GPS and Plantower G7003S were defined when using Arduino Mega 2560‘s universal asynchronous receiver/transmitter (UART) ports, which are only equipped with three pairs, which means the extensibility is limited. Although it is possible to transfer data using multiple Mega board communication, it is complex and easy to lose data. Moreover, SBEDAD was initially set up as a monolithic device that reads and stores data locally to ensure the time-resolution granularity of the data, reduce data loss and frame loss, and ensure the security of the data during transmission. SBEDAD is not equipped with any mobile data transfer module or networking module, which means it does not have real-time feedback capabilities. In future research, cloud-based data networking may be attempted, which will lead to some new technical challenges, such as data transfer protocols, data collaboration, edge computing, and so on.

In the original design, we had hoped to provide feedback on the quality of the road surface and add jitter body sensing detection. However, because the ceramic bottom jitter sensor was damaged in the field research and generated frequent error data, the jitter sensor could not effectively provide feedback on the jitter experience while riding due to the time-limited feedback and the limited range and accuracy. It is expected that an IMU system will be added to future experiments to provide pilot feedback on bumps caused by road surface quality in urban environments.

Another future research plan is to apply a Python script to create a customer-friendly interface for SBEDAD data reading, which will synthesize image frame extraction with latitude and longitude data acquisition, sensor data cleaning, and final tabulation of valid data and waveform graph visualization. This is to facilitate its use by urban designers, outdoor built environment designers, and even indoor built environment designers who do not have basic knowledge of programming and to provide designers with new perspectives on the data to improve their attention to the environment. Based on the results of this research, we hope to liberate the shackles of data on urban researchers and managers, ignite more curiosity, and cultivate an enduring passion for cross-disciplinary urban research, thereby realizing accurate urban intelligence.

## Figures and Tables

**Figure 1 sensors-24-03096-f001:**
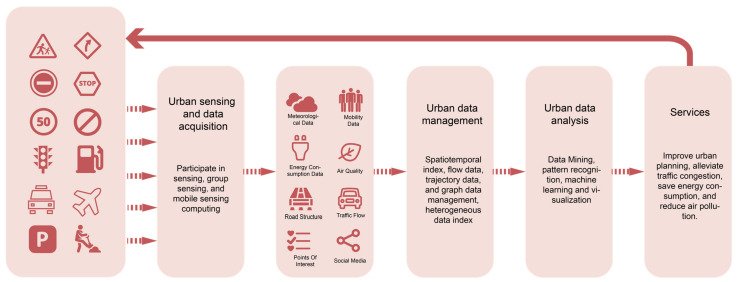
Street built environment data platform based on sensors and IoT technology to realize urban sensing and data acquisition; organize and obtain a variety of data, such as human mobility data, social media data, etc.; and provide services for the healthy development of the city through urban data management and urban data analysis.

**Figure 2 sensors-24-03096-f002:**
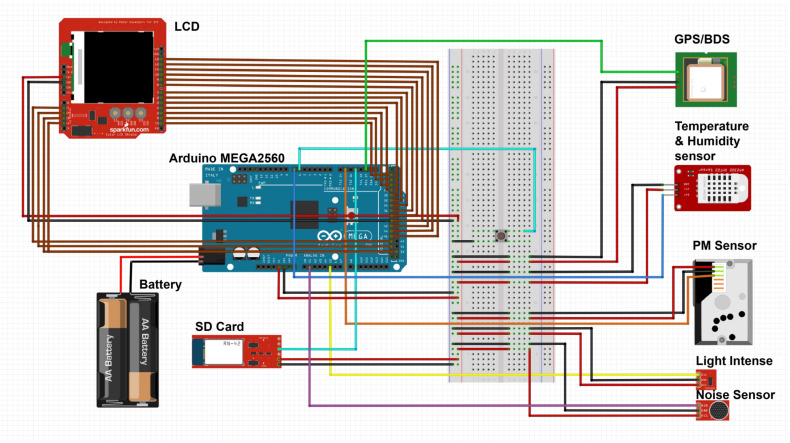
Electronic circuit of the SBEDAD. The Arduino built-in UART enables two devices to communicate at a time, so SBEDAD has the GPS and PM sensors connected to the serial port. The DHT22 data are collected at the digital pin, while the BH1750 and noise sensor data are collected at the analog pins. The availability of each component is confirmed in the overall function realization.

**Figure 3 sensors-24-03096-f003:**
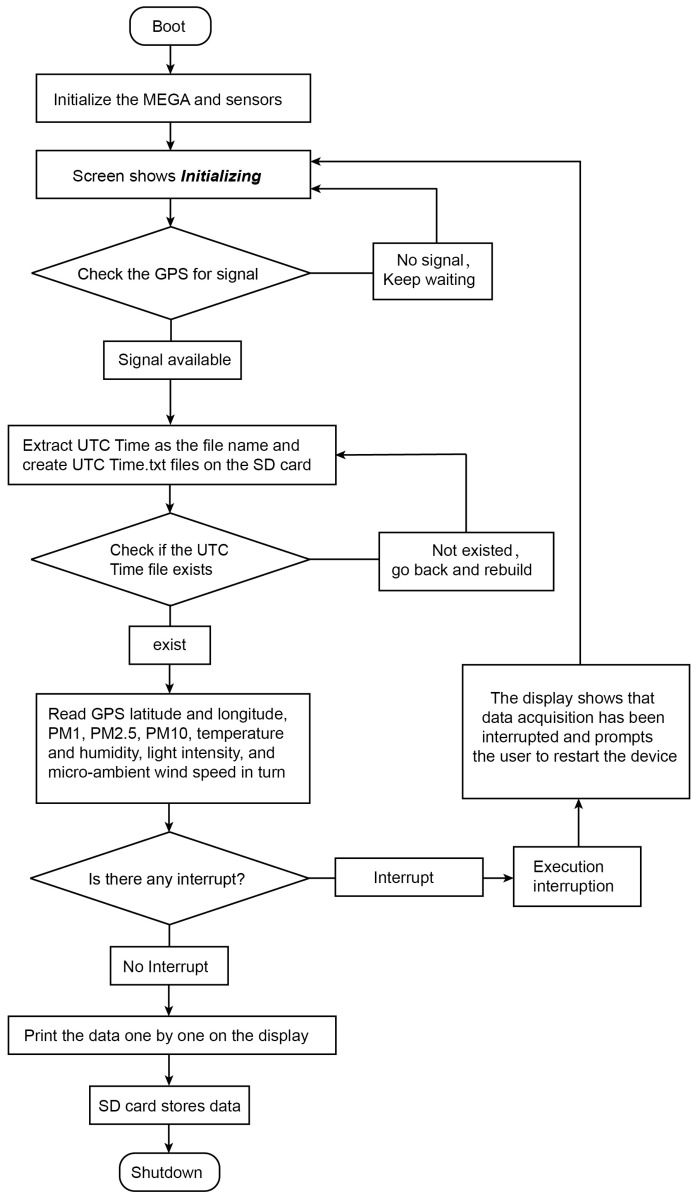
Flow chart of SBEDAD.

**Figure 4 sensors-24-03096-f004:**
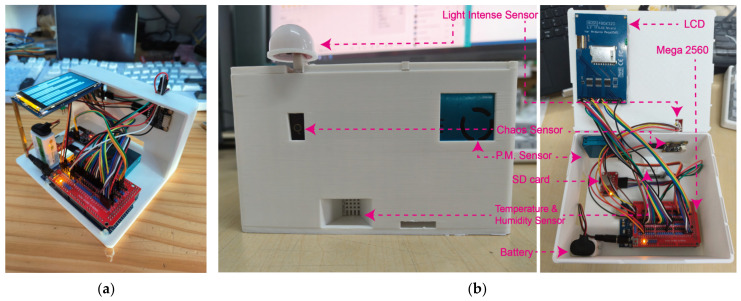
(**a**) The 1st version SBEDAD, half-shelled, with TEMT6000 as the light intensity sensor; the screen is printing the sensing data to ensure the users get the running status of the equipment. (**b**) SBEDAD with PLA shell (2nd version) using BH1750 as the light intensity sensor.

**Figure 5 sensors-24-03096-f005:**
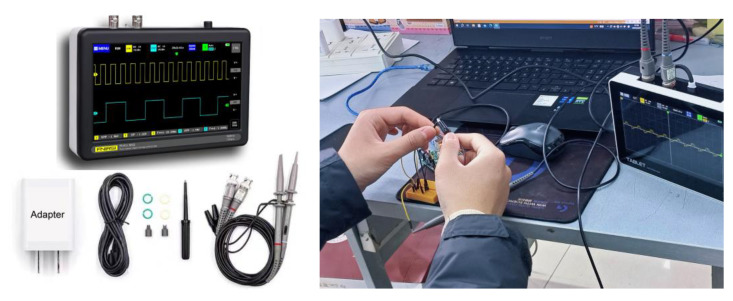
Evaluation calibrations of SBEDAD using the digital oscilloscope FNIRSI-1013D.

**Figure 6 sensors-24-03096-f006:**
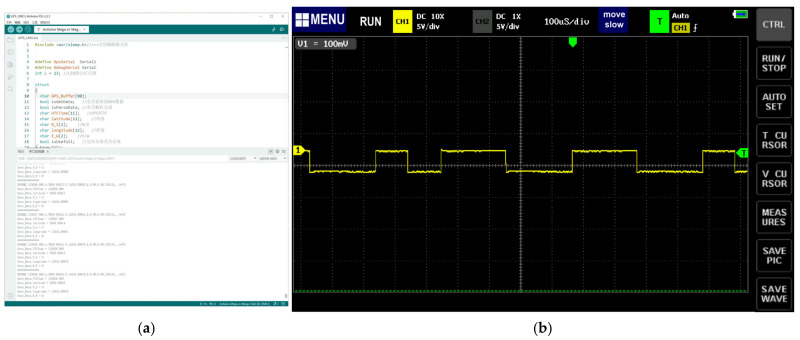
(**a**) Arduino IDE reads GPS/BDS data via a serial port. (**b**) Measurement of the real frequency of GPS/BDS sensor.

**Figure 7 sensors-24-03096-f007:**
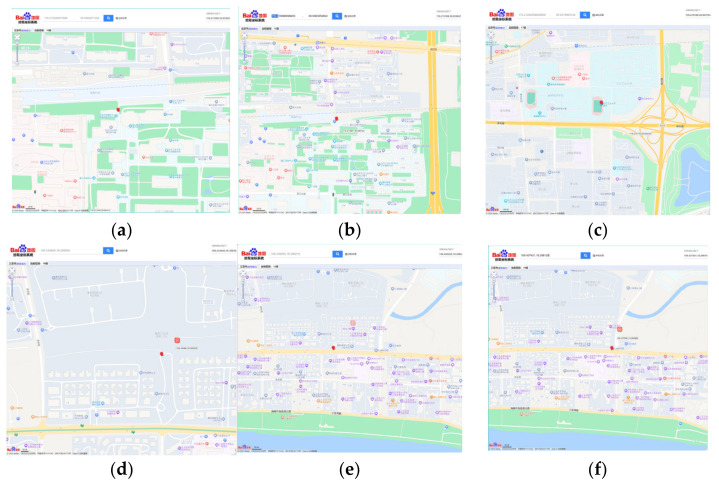
Field test of GPS/BDS sensor in Beijing (**a**–**c**) and Sanya (**d**–**f**). By inverting the geographic location by latitude and longitude on the Baidu map (upper left column), the result (the red icons) showed that ATGMH-5N has high accuracy and meets the urban spatial data requirements.

**Figure 8 sensors-24-03096-f008:**
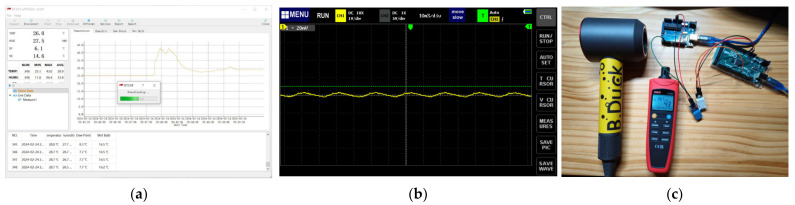
Lab test of the temperature and humidity sensor. (**a**) Temperature and humidity data for UT332+ through the host computer application. (**b**) Oscillograph of the DHT22. (**c**) Testing the sensor’s ability to respond quickly to changes in temperature and humidity using a hairdryer.

**Figure 9 sensors-24-03096-f009:**
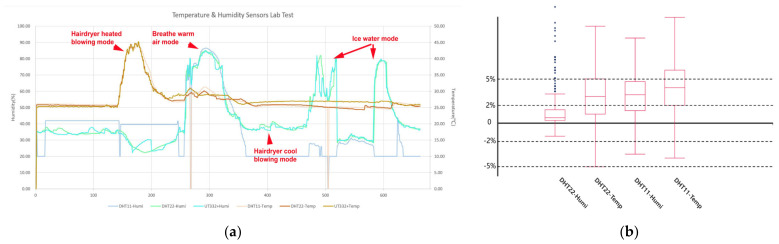
(**a**) The 600-set data comparison of DHT11, DHT22, and UT332+ in different modes. (**b**) Statistical analyses of the error rate between the sensors and professional tester data for the 600 sets of data.

**Figure 10 sensors-24-03096-f010:**
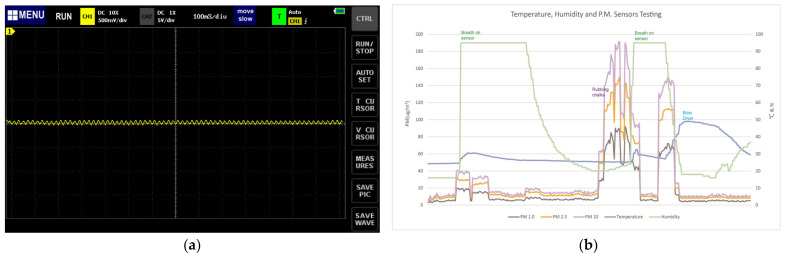
(**a**) Measurement of the real frequency of Plantower G7003s. (**b**) The lab test with the Plantower G7003S using sandpaper to rub nails, plastics, and chalks.

**Figure 11 sensors-24-03096-f011:**
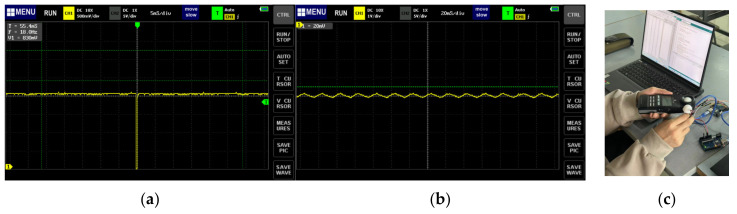
Measurement of the real frequencies of TEMT6000 (**a**) and BH1750 (**b**). Lab test of TEMT6000, BH1750, and Konica T-10A (**c**).

**Figure 12 sensors-24-03096-f012:**
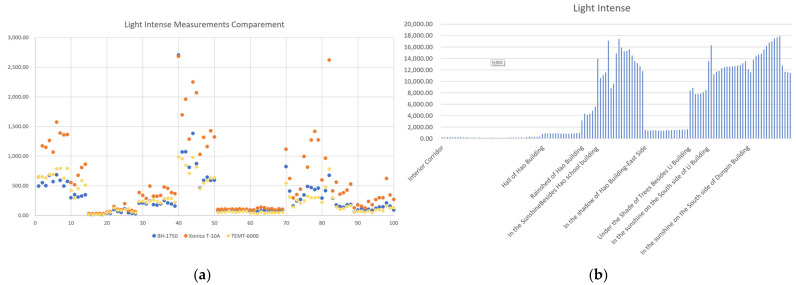
(**a**) Lab test of light intensity with TEMT6000, BH1750, and Konica T-10A. The TEMT6000 shows accuracy in limited light environments (green dots), but in environments with high luminous flux, the results deviate greatly from professional measuring instruments (orange dots). (**b**) Field test of light intensity with BH1750 (**b**).

**Figure 13 sensors-24-03096-f013:**
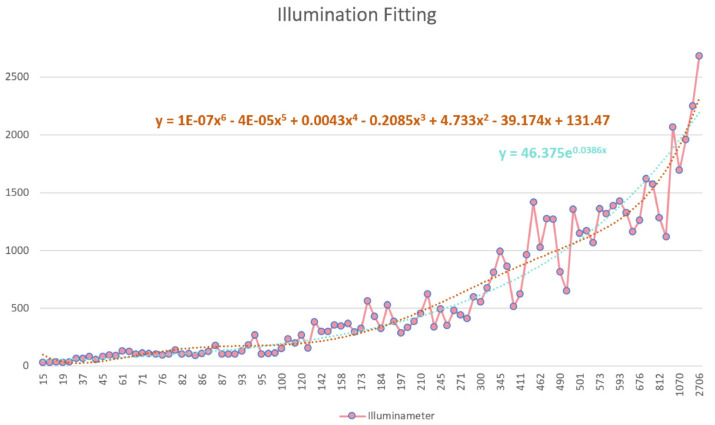
Illumination fitting of BH1750. A total of 100 sets of data in a daily light environment were collected (pink dots), and the data from BH1750 were fit to T-10A to perform data calibration for BH1750 in two ways, the fitting curve in brown is the result of Equation (1), while fitting curve in green is the result of Equation (2).

**Figure 14 sensors-24-03096-f014:**
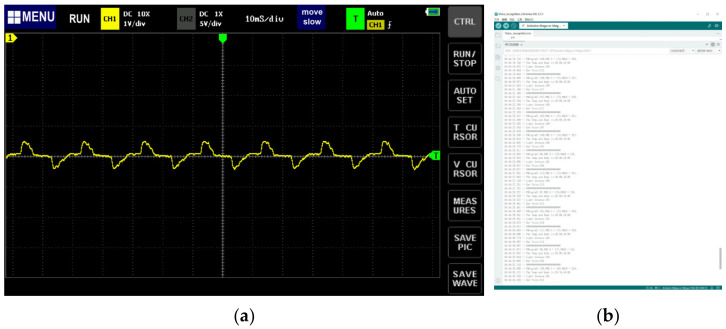
(**a**) Measurements of the real frequency of the Yahboom sensor. (**b**) We obtained environmental noise in the lab by playing different types of sounds.

**Figure 15 sensors-24-03096-f015:**
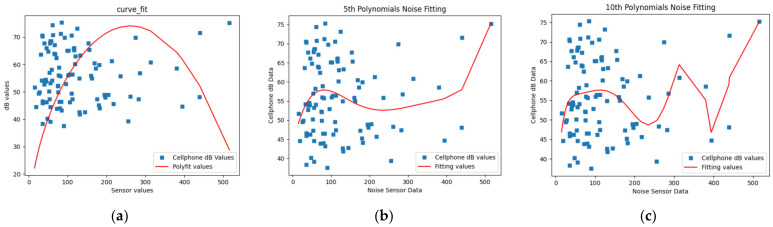
Data fitting with the Yahboom sensor and cellphone sensor. We fit with three modes: (**a**) curves, (**b**) 5th degree polynomials, and (**c**) 10th degree polynomials to the data using a Python script.

**Figure 16 sensors-24-03096-f016:**
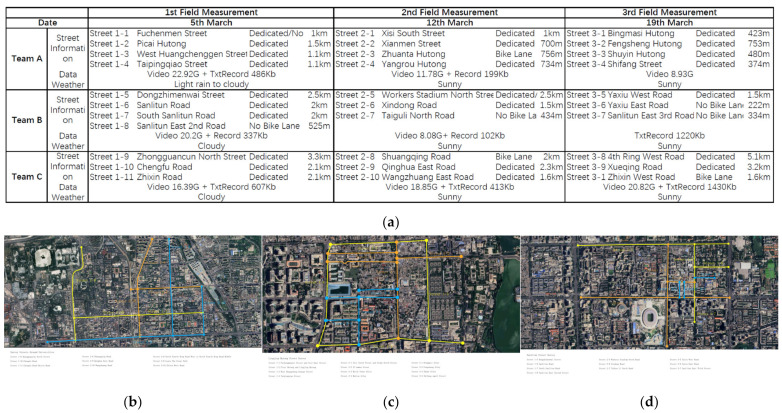
Field research with SBEDAD in three major areas in the Beijing core area. (**a**) Statistical information of field measurements using SBEDAD. (**b**–**d**) The streets being measured by three teams.

**Figure 17 sensors-24-03096-f017:**
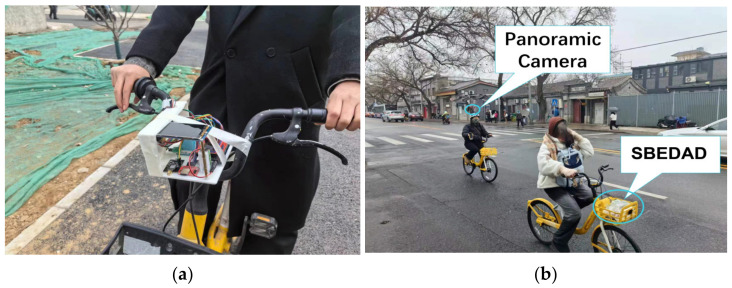
Field test with SBEDAD. (**a**) The first version of SBEDAD. (**b**) The measurements used both the panoramic camera and SBEDAD.

**Figure 18 sensors-24-03096-f018:**
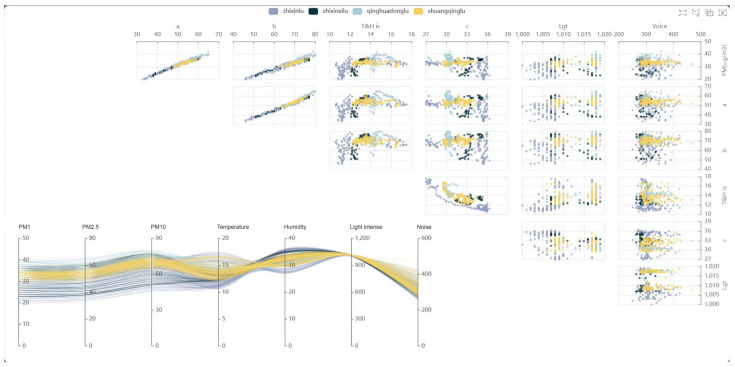
Urban spatial data visualization by Urban Design Studio students using the data collected from SBEDAD in conjunction with the D3.js visualization script to visualize and analyze the spatial–temporal features of urban sensing data to deeply reveal the features of human activities in the urban physical built environment.

**Figure 19 sensors-24-03096-f019:**
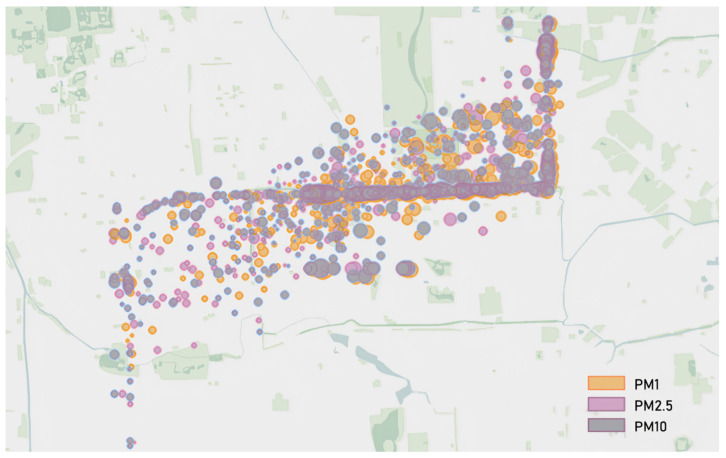
Urban spatial data visualization by Urban Design Studio students using the data collected from SBEDAD in conjunction with the Tableau to visualize and analyze the spatial–temporal features of PM concentration.

**Table 1 sensors-24-03096-t001:** Technical parameters of the Arduino Mega 25.60 [40].

Mega 2560 Specification Parameter	
Operating voltage	5 V
Input voltage (recommended)	7–12 V
Input voltage (limit)	6–20 V
Digital I/O pins	54
Analog I/O pins	16
DC I/O pin	40 mA
DC for 3.3 V pin	50 mA
Flash memory	256 KB
SRAM	8 KB
EEPROM	4 KB
Clock frequency	16 MHz

**Table 2 sensors-24-03096-t002:** Configuration of SBEDAD.

Purpose	Type	Model	Response Time
Printed circuit board	Arduino	MEGA2560	--
Longitude and latitude	GPS and BDS	ATGM336H-5N	0.1 ms
Temperature and humidity	Humidity and temperature	DHT22	10 ms
Air pollution	PM	Plantower G7003S	100 ms
Light	Light sensor	TEMT6000	20 ms
	Light sensor	BH1750	5 ms
Noise	Voice sensor	Croco-Sound	20 ms
Air flow	Air flow sensor	Air flow sensor	10 ms
Status display	LCD	HX8357-B	
Data storage	SD	OV7670	
Video record	Panoramic camera	Pilot One	

**Table 3 sensors-24-03096-t003:** Price and features of SBEDAD.

Model	Nature	Commercial	Price
MEGA2560	Obligatory	YES	35.28
ATGM336H-5N	Obligatory	YES	10.93
DHT22	Obligatory	YES	1.08
Plantower G7003S	Obligatory	YES	12.18
TEMT6000	Obligatory	YES	0.90
BH1750	Obligatory	YES	1.1
Croco Sound	Obligatory	YES	1.37
Air flow sensor	Optional	YES	6.64
HX8357-B	Obligatory	YES	1.38
OV7670	Obligatory	YES	0.48
PTQS1005	Optional	YES	69.16
Heart beat rate sensor	Optional	YES	1.51
3D-printed shell	Optional	No	5
Batteries	Obligatory	YES	2.77
Cable and welding	Obligatory	No	1
Switching button	Optional	YES	1.11
Resistor	Obligatory	YES	0.1

**Table 4 sensors-24-03096-t004:** Accuracy of data acquired by each sensor of SBEDAD.

Model	Range	Error Precision	SBEDAD Accuracy
Temperature	18–25 °C [44,45]	±1 °C	±0.5 °C
Humidity	40–70% [44,45]	>1%	±1%
PM	0–500 μg/m^3^ [46]	±25%	±10–100 μg/m^3^
Light	10,000–18,000 Lux [47]	±20%	>10 lux
Sound	40–120 db [48]	±5 db	0–40 m/s
GPS	--	--	2 m

**Table 5 sensors-24-03096-t005:** GPS and BDS model.

Model	Nature
Manufacturer	Yahboom
Cold-start acquisition sensitivity	−148 dBm
Hot-start acquisition sensitivity	−156 dBm
Recapture sensitivity	−160 dBm
Tracking sensitivity	−162 dBm
Positioning accuracy	2.5 m(CEP50)
Typical power consumption	25 mA @3.3 V
Agreement	NMEA0183
Maximum height	8000 m
Maximum speed	515 m/s
Operating temperature	−40 °C to + 85 °C
Size and weight	10.1 mm × 9.7 mm × 2.4 mm, 0.6 g

**Table 6 sensors-24-03096-t006:** Temperature and humidity model.

	DHT11	DHT22	UNI-T UT332+
	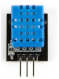	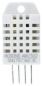	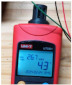
Temperature range	0 to 50 °C +/−2 °C	−40 to 80 °C +/−0.5 °C	−20 to 80 °C
Humidity range	20 to 90% +/−5%	0 to 100% +/−2%	0 to 100%
Resolution	Humidity: 1%Temperature: 1 °C	Humidity: 0.1%Temperature: 0.1 °C	Humidity: 0.1%Temperature: 0.1 °C
Operating voltage	3–5.5 V DC	3–6 V DC	3–5 V DC
Current supply	0.5–2.5 mA	1–1.5 mA	0–10 A
Sampling period	1 s	2 s	1 s
Price	USD 1 to 5	USD 4 to 10	USD 60 to 80

**Table 7 sensors-24-03096-t007:** Plantower G7003S model.

Parameter	Indicator	Unit
Principle	Laser Scattering	---
Measuring range of particles	0.3~1.0; 1.0~2.5; 2.5~10	µm
Effective range of concentration	0~500	µg/m^3^
Maximum mass	≥1000	µg/m^3^
Mass concentration resolution	1	µg/m^3^
DC voltage	4.5–5.5	V
Working current	≤100	mA
Standby current	≤200	µA
operating temperature range	−10~+60	°C
Operating humidity range	0~99%	V

**Table 8 sensors-24-03096-t008:** Light intensity model.

	TEMT6000	BH1750	Konica T-10A
	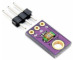	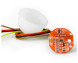	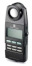
Intense range	0–1023	1–100,000 lx	0.01–299,900 lx
Deviation	1	1 lx	0.01 lux
Operating voltage	1.5–6 V DC	1.8–4.5 V DC	1.5 V DC
Operating temp	−10–100 °C	−10–85 °C	−20–55 °C
Current supply	20 mA	0.5–2.5 mA	400 mA
Sampling period	5 ms	20 ms	28 ms
Power dissipation	100 mw	260 mw	244 mw
Price	USD 1	USD 3	USD 2000

**Table 9 sensors-24-03096-t009:** Environmental noise model.

	Yahboom	Noise App
	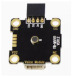	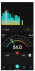
Intense range	0–1023	0–140 dB
Deviation	±1%	0.1 dB
Operating voltage	3.3–5 V DC	0.8–1 V DC
Operating temperature	−40–100 °C	−10–85 °C
Current supply	20 mA	80 mA
Sampling period	20 ms	1000 ms
Power dissipation	100 mw	18–26 W
Price	USD 0.9	USD 12

## Data Availability

Data unavailable due to privacy restrictions.

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
