# Peer review of "Portable Arduino-Based Multi-Sensor Device (SBEDAD): Measuring the Built Environment in Street Cycling Spaces"

_sensors, 2024, doi:10.3390/s24103096_

Round 1

Reviewer 1 Report

Comments and Suggestions for Authors

This study piques my curiosity because of its potential to built environment measurements of street cycling spaces in IoT through its aims and approach. What ignited my interest in the article was its main argument. The writing is straightforward and easy to understand, and it provides strong evidence to support its core claims. If the authors are willing to implement the adjustments I suggested, I would be pleased to review the work again. I really hope that by making these changes, the article will become more organized and easier to understand.

1)    Update the abstract- Start with Background, scope, and problem definition in first 5 lines. Then add what is the aim/objective of the paper, and then add 2 line what is proposed and then add 2 lines what novelty is associated with the paper. And then in last three lines, highlight in what %age and in what parameters the proposed methodology is better as compared to existing techniques and what is the overall analysis.

2)    The part of system requirement and design study should lists many relevant studies, and I think an overall summary and analysis will Enable readers to better understand existing research.

3)    Elaborate about what overall technical gaps are observed in existing works, that led to the design of proposed methodology.

4)    Elaborate everything with proper theory, Tables with Data on the basis of which graphs are plotted and then add diagrams in Functional Validation section.

5)    Discuss limitations of proposed SBEDAD.

6)    Although the article provided a comprehensive analysis of the proposed method, additional details regarding the assumptions and constraints used would have been beneficial. Is there anything about the suggested approach that makes it hard to implement in actual Internet of Things deployments?

7)    The conclusion section may be rewritten to be clearer and to highlight more specific reasons why this strategy is superior to others.

8)    Figs are mostly coarse grained, use pdf as source file.

9)    A proof-read is needed to fix minors and typos as well as to enhance the effectiveness of key statements of abstract, intro and conclusion

10) The values given to the setting options need to be discussed and motivated (e.g., does it refer to a particular use case?).

Comments on the Quality of English Language

 A proof-read is needed to fix minors and typos as well as to enhance the effectiveness of key statements of abstract, intro and conclusion

Reviewer 2 Report

Comments and Suggestions for Authors

The paper is well written and is based on thorough measurements, analysis and research, but a few improvements should be performed.

1. In the abstract, the authors mention that they are using a 6K panoramic camera for multi-source data acquisition, which is also mentioned in section 2.1 of the article. However, at page 8 of the paper, authors mention a "8K panoramic video camera PilotOne". Is the camera mentioned at page 8 the only one used in the experiments and it is the same as the one in the abstract of the article? If yes, the inconsistency in the actual resolution of the camera should be corrected in the text.

2. Minor inconsistencies in the name and numbering of tables should be corrected. At page 5, the name of Table 2 is missing: "Table 2. This is a table. Tables should be placed in the main text near the first time they are cited." The table at page 12 presents data about light intensity sensors, but the name of the table is "Temperature and Humidity Model" and, also, the number of this table and the following tables should be corrected.

3. Figure 16 presents a great amount of field measurement data. The term "video" in the table included in figure 16 is misspelt. The device proposed by the authors collects lots of gigabytes of video data from the camera. In my opinion, the paper doesn't explain very well why this huge amount of video data is important for the experiment, how relevant data is extracted from the videos and if this video data is compressed or how it is stored for further analysis. Are there any means to reduce the size occupied by this video data on the drives? Is there any algorithm which parses automatically the frames of the videos?

4. The conclusions section of the article should be extended and the authors should stress out in a more detailed way which is the connection between each set of experimental data obtained and the conclusions of their work and how this data can be used for further research directions.

Comments on the Quality of English Language

In the abstract of the article, the authors mention that their proposed device is used "for built environment measurements of street cycling spaces". In my opinion, the terms "built environment measurements" should be further explained in order to better understand what the built environment is. Is it referring only to dense urban areas, is it referring to outdoor environments or also to indoor environments?

At the end of section 4 of the article, the authors mention: "Simply interpreting a city through one type of data probability is not comprehensive". I suppose that instead of "probability", the term "probably" should have been used, because the authors intended to mention that one type of data is not enough in order to analyze the complexity of the city.

Round 2

Reviewer 1 Report

Comments and Suggestions for Authors

author has addressed all recommended comments to improve the quality of the paper